# Subclinical Hypocalcemia Across Lactation Stages Reflects Potential Metabolic Vulnerability in Korean Holstein Cows

**DOI:** 10.3390/vetsci12050495

**Published:** 2025-05-19

**Authors:** Hector M. Espiritu, Md. Shohel Al Faruk, Hee-Woon Lee, Jaylord M. Pioquinto, Sang-Suk Lee, Yong-Il Cho

**Affiliations:** 1Department of Animal Science and Technology, Sunchon National University, Suncheon 57922, Jeonnam, Republic of Korea; hectorme@scnu.ac.kr (H.M.E.); shohel@cvasu.ac.bd (M.S.A.F.); jmpioquinto@clsu.edu.ph (J.M.P.); rumen@scnu.ac.kr (S.-S.L.); 2Department of Physiology, Biochemistry and Pharmacology, Chattogram Veterinary and Animal Sciences University, Khulshi, Chattogram 4225, Bangladesh; 3Mari Animal Medical Center, Baegan-Myeon, Yongin 17178, Gyeonggi, Republic of Korea; iamlhw@naver.com

**Keywords:** subclinical hypocalcemia, dairy cows, lactation stages, serum biomarkers, parity

## Abstract

Some dairy cows develop low blood calcium without showing obvious signs of illness. This hidden condition, called subclinical hypocalcemia (SCH), is easy to miss but may affect health. We studied 859 cows from 49 farms in South Korea to see how common SCH is during lactation. About 1 in 10 cows were affected, and over two-thirds of farms had at least one case, showing that SCH is widespread. Older cows, especially in their fourth lactation or more, were over three times as likely to be affected. Even though milk production was similar between affected and healthy cows, blood mid-lactation showed signs of internal stress in cows with SCH, including higher phosphorus and lower levels of proteins and fats made by the liver. These findings suggest that regular blood monitoring during lactation can help detect hidden issues and improve herd health.

## 1. Introduction

Subclinical hypocalcemia (SCH) is a common metabolic disorder in high-producing dairy cows and has historically been associated with the transition period, the critical window surrounding parturition, and early lactation [1,2,3]. Defined as a reduction in blood calcium (Ca) concentration without overt clinical signs, SCH has been linked to impaired neuromuscular function, suppressed immune responses, increased disease susceptibility, reduced fertility, and significant loss in productivity [2,4,5]. SCH also disrupts energy metabolism and increases disease risk throughout lactation [4,5,6]. Recent findings further show altered neutrophil function linked to SCH-associated metabolic shifts [7]. Its high prevalence, affecting up to 50% of multiparous cows during early lactation, and its hidden economic costs, estimated to be around USD 70 per affected cow, underscore its significance for dairy herd health management [1,4,8]. Due to the steep rise in Ca demand for colostrum and milk synthesis, most research on SCH has focused exclusively on the first 24–72 h postpartum, when the condition is presumed to be most acute [9,10,11]. The postpartum period represents a critical window of metabolic vulnerability in dairy cows, during which profound physiological adaptations occur to meet the demands of lactation. This phase is characterized by a heightened risk of metabolic disorders such as subclinical ketosis, negative energy balance (NEB), and systemic inflammatory responses [12,13]. Within this complex physiological context, SCH may emerge as a key, but often underrecognized, component of postpartum metabolic instability.

However, this traditional view may underrepresent the physiological burden of Ca instability in modern dairy systems. Holstein cows produce up to 30–40 kg of milk per day, with some exceeding 60 kg under optimal conditions, significantly increasing Ca demand during lactation [2,14]. This high metabolic output sustained across extended lactation cycles implies that Ca homeostasis may remain vulnerable beyond the transition phase [15,16]. Moreover, parity-associated changes in Ca metabolism, especially in older cows, may contribute to chronic or late-emerging deficiencies that have not been captured in studies focused on early days in milk (DIM) [3,11].

In this context, applying the SCH diagnostic threshold (serum Ca ≤ 8.2 mg/dL) beyond the immediate postpartum period offers a broader and potentially more sensitive lens to detect latent or sustained metabolic stress in otherwise clinically NCC. Although this approach extends the classical definition of SCH, it reflects growing recognition that subclinical Ca imbalances may persist or recur outside the transition window. Such conditions, even in the absence of overt disease, may signal metabolic inefficiency, reduced adaptive capacity in older cows, or subtle immune modulation, all of which could affect long-term health and performance [3,16,17,18]. Identifying these patterns may allow for earlier intervention and more tailored herd management strategies [2,19].

To explore this possibility, we conducted a nationwide cross-sectional study involving 859 lactating Holstein cows from 49 commercial dairy farms across five regions of the Republic of Korea. We assessed serum Ca concentrations alongside blood- and milk-based physiological indicators, including complete blood counts, serum biochemistry, and milk production traits. Unlike previous studies that focused primarily on early lactation or fresh cows, we included animals from all lactation stages and parities to better reflect real farm diversity and conditions. We hypothesized that SCH is not confined to the early postpartum period and that cows at various lactation stages and parities may exhibit persistent or intermittent low Ca states, accompanied by distinct changes in blood, serum, and milk parameters. This study challenges the conventional view that SCH is relevant only during the transition period and supports a broader physiological interpretation of Ca imbalance throughout lactation.

## 2. Materials and Methods

### 2.1. Ethics Statement

This study was conducted as part of routine dairy monitoring and adhered to the Animal Protection Act of the Republic of Korea and the relevant animal welfare guidelines. All procedures, including blood collection, were performed by licensed veterinarians to ensure proper animal handling and to minimize stress. Farm owners provided informed consent prior to participation, and no experimental interventions were applied beyond routine health assessments.

### 2.2. Study Population and Classification

A total of 859 lactating Holstein cows from 49 dairy herds across five regions of Korea (Chungcheong, Gyeonggi, Gyeongsang, Jeolla, and Gangwon) were included in the final analysis. Farms were selected through veterinary contact networks, including 15 farms each from Chungcheong and Gyeonggi, 9 from Gyeongsang, and 5 from Jeolla and Gangwon. This cross-sectional study included all eligible cows with complete data sampled once during scheduled herd visits. No a priori power analysis was conducted due to the observational design and the lack of a single predefined primary outcome. The sample size reflects a pragmatic cross-sectional approach intended to maximize field representativeness. All cows were housed in free-stall barns and fed total mixed rations (TMRs) formulated according to Korean dairy feeding standards. The use of anionic salts, oral Ca boluses, or postpartum interventions was not systematically recorded. Thus, this study reflects the metabolic status of cows managed under typical commercial conditions.

Cows were randomly selected and classified into two groups based on serum total Ca concentration: normocalcemic (NCC; ≥8.3 mg/dL) and subclinical hypocalcemic (SCH; ≤8.2 mg/dL or ≤2.05 mmol/L, without clinical signs). The threshold was chosen to improve diagnostic specificity beyond the transition period and aligns with the lower bound of the physiological reference range for adult dairy cows. Although higher cutoffs such as 8.5 mg/dL are often used for early postpartum cows, our inclusion of cows across all lactation stages and the cross-sectional study design warranted the use of a stricter threshold.

Lactation stages were classified based on DIM as follows: early (0–49 DIM), high production (50–109 DIM), mid-lactation (110–219 DIM), and late lactation (≥220 DIM), or as provided by the collaborating milk testing laboratory. Additional recorded variables included parity (1, 2, 3, or ≥4), milk yield, composition, CBC, and serum biochemical parameters. Milk composition (fat, protein, and SNF content) was analyzed by a third-party laboratory as part of routine herd testing and shared with the collaborating veterinary clinic. The clinic also recorded individual-level data, including parity, age, and lactation stage, at the time of sampling.

### 2.3. Blood Sampling

Blood samples were collected from the coccygeal vein using a 19-gauge, 3.8 cm needle and a 10 mL syringe. Two types of collection tubes were used: a 5 mL Serum Separator Tube (SST^TM^ II Advance, BD Vacutainer^®^, Wayne, NJ, USA) for serum analysis and a 3 mL K_2_EDTA Tube (BD Vacutainer^®^, Wayne, NJ, USA) for whole blood analysis. Immediately after collection, blood samples were stored in portable coolers maintained at ~4 °C. Samples for serum analysis were centrifuged at 4000 rpm for 10 min. Following separation, serum samples were aliquoted and stored at −20 °C until laboratory analysis. EDTA-anticoagulated whole blood samples for CBC analysis were also stored at ~4 °C and processed within 24 h of collection. This handling protocol followed validated standards for maintaining the stability of hematological and biochemical parameters in bovine blood [20].

### 2.4. Complete Blood Count (CBC) Analysis

CBC was performed using an automated hematology analyzer (IDEXX ProCyte Dx^TM^, Westbrook, ME, USA). The evaluated parameters included red blood cell (RBC) count, hematocrit (HCT), hemoglobin (HGB), mean corpuscular volume (MCV), mean corpuscular hemoglobin (MCH), mean corpuscular hemoglobin concentration (MCHC), total white blood cell (WBC) count, and differential leukocyte count (neutrophils, lymphocytes, monocytes, eosinophils, and basophils).

### 2.5. Serum Analysis

Serum biochemical parameters were analyzed using an automated serum chemistry analyzer (IDEXX Catalyst One^TM^, Westbrook, ME, USA) to measure 12 biochemical indices, including Ca, phosphorus (P), magnesium (Mg), blood urea nitrogen (BUN), total protein (TP), albumin (ALB), globulin (GLB), ALB/GLB ratio, aspartate aminotransferase (AST), gamma-glutamyl transferase (GGT), total bilirubin (TBIL), and cholesterol (CHOL). Additionally, glucose and ketone (β-hydroxybutyrate; BHBA) levels were measured using a portable glucose and ketone test meter (FreeStyle Optium Neo H, Abbott Diabetes Care Ltd., Seoul, Republic of Korea) with Precision Xtra glucose and ketone test strips.

### 2.6. Statistical Analysis

Statistical analyses were conducted using IBM SPSSv26 and R (version 4.3.3; R Core Team, Vienna, Austria) The prevalence of SCH across lactation stages and parity groups was compared using the chi-squared test. Statistically significant differences were explored using pairwise comparisons with letter groupings to identify the between-group differences. Independent *t*-tests were used to compare milk yield, composition, CBC, and serum biochemical parameters of SCH and NCC at each parity and lactation stage. To control the risk of Type I error, a two-way ANOVA was performed to evaluate the main effects and interactions between SCH status and either parity or lactation stage on milk production traits, followed by post-hoc comparisons using Tukey’s honest significant difference (HSD) test. All analyses were based on planned hypotheses regarding group differences, and interpretation accounted for the number of comparisons conducted. Group-level differences were annotated using lowercase letters (a, b, c) to indicate differences among parity or lactation stages and uppercase letters (X, Y) to indicate differences between SCH and NCC. Statistical significance was set at *p* ≤ 0.05.

## 3. Results

### 3.1. Prevalence and Severity of Subclinical Hypocalcemia (SCH)

The overall prevalence of SCH among lactating cows was 9.4% (81/859). At the herd level, 69.4% of the farms (34 out of 49) had at least one SCH-positive cow, indicating that SCH is widespread in Korean dairy herds despite a relatively low individual-level prevalence. The prevalence of SCH increased progressively with parity, from 4.5% in cows with parity 1 to 15.7% in cows with parity 4 or higher, which was statistically significant (*p* < 0.05; Figure 1A). Pairwise comparisons showed that cows at parity ≥ 4 had a significantly higher prevalence than those at parity 1 (*p* < 0.05), whereas the differences among intermediate parities were not significant. The prevalence of SCH was highest in early lactation (14.5%), followed by the late (8.6%), high (7.9%), and mid (6.5%) stages. However, these differences were not significant (Figure 1B).

Serum Ca concentrations were compared exclusively among SCH-positive cows to evaluate the severity of SCH. No statistically significant differences in serum Ca levels were observed across the parity groups (Figure 1C). In contrast, serum Ca levels varied significantly by lactation stage (*p* < 0.05), with early lactation cows (7.86 ± 0.32 mg/dL; 1.96 ± 0.08 mmol/L) exhibiting significantly lower Ca concentrations than those in late lactation (8.11 ± 0.27 mg/dL; 2.02 ± 0.07 mmol/L). Cows in the high and mid stages showed intermediate values (Figure 1D).

### 3.2. Milk Yield and Composition

Milk production parameters did not differ significantly between cows diagnosed with SCH and their normal counterparts. The average milk yield was 37.87 kg for SCH cows and 37.86 kg for NCC (*p* > 0.05). Similarly, no significant differences were observed in milk fat (3.82% vs. 3.93%), protein (3.21% vs. 3.15%), or SNF (8.79% vs. 8.84%) contents between the two groups (Figure 2).

### 3.3. Milk Yield and Composition by Parity

Using independent *t*-tests within each parity group, no statistically significant differences were observed in milk yield or composition between cows with SCH and their normal counterparts. However, milk yield was numerically comparable between SCH and NCC at certain parities. For example, at parity 3, SCH cows yielded 38.71 kg/day compared to 42.34 kg/day for NCC, and at parity ≥ 4, 38.99 kg/day in SCH cows versus 41.48 kg/day in NCC (Figure 3A). Regarding milk fat, SCH cows at parity 2 had a lower average (3.20%) than NCC (3.89%). However, the difference was not statistically significant (Figure 3B). Protein content was similar across all parities, with slightly higher levels in SCH cows at parity 3, but with no significant differences between SCH and NCC (Figure 3C). The same was true for SNF content, with nearly identical averages observed between SCH and NCC at each parity (Figure 3D).

To explore broader trends, a two-way ANOVA was used to evaluate the effects of parity and SCH status on milk traits. Milk yield increased from parity 1 to 3, followed by a slight decline in parity ≥ 4. The highest average was recorded in NCC cows at parity 3 (42.34 kg/day) and the lowest in NCC at parity 1 (32.21 kg/day). Among SCH cows, values were relatively stable across parities 2 to 4+ (38.71–39.47 kg/day), with slightly lower levels in parity 1 (32.98 kg/day). However, these differences were not statistically significant (Figure 3A). Although yield trends across parity were observed, statistically significant differences were limited to specific traits such as milk fat and SNF.

Milk fat ranged from 3.20% to 4.00%, with the lowest observed in SCH cows at parity 2 and the highest in NCC at parity 1. SCH cows at parity 2 had significantly lower fat percentages than NCC cows at parities 1 and 2 (Figure 3B). Milk protein ranged from 3.05% to 3.30%, with slightly higher values at parity 3, though the differences were not statistically significant (Figure 3C). SNF content gradually declined with increasing parity.

There were higher values in NCC at parity 1 (8.90%) and parity 2 (8.86%), and the lowest in NCC was at parity ≥ 4 (8.68%). NCC at parity 1 and 2 had significantly higher SNF values than did parity ≥ 4 cows (Figure 3D).

### 3.4. Milk Yield and Composition by Lactation Stage

No statistically significant differences were observed in milk yield or composition between SCH and NCC during the individual lactation stages, except for protein in the early stage, when SCH cows showed higher levels than NCC (Figure 4C). At other stages, SCH and NCC exhibited comparable milk yield and fat, protein, and SNF content (Figure 4A,B,D).

Significant differences were observed for all milk traits across the lactation stages. The difference in milk yield was most pronounced between SCH cows in the high stage, which had the highest average, and NCC in the late stage, which had the lowest average (Figure 4A). Milk fat also varied significantly by stage, with the highest value observed in SCH cows during the early stage and the lowest value observed in SCH cows during the high stage (Figure 4B). The highest level of milk protein was found in SCH cows during the late stage, whereas the lowest level was found in NCC during the early stage (Figure 4C). The SNF content showed a similar pattern, with the highest value in late-stage cows and the lowest in SCH cows during the high stage (Figure 4D).

### 3.5. Complete Blood Count (CBC)

All CBC parameters were within the expected physiological reference ranges in both the SCH and NCC, and no statistically significant differences were observed between the groups (Table 1). Although CBC values remained within normal limits, slight downward trends in RBC and WBC counts in SCH cows may still reflect early or compensated physiological stress, which is consistent with a subclinical homeostatic challenge. RBC count, HCT, and HGB levels tended to be slightly lower in SCH cows than in NCC, although these values remained within the normal range. Erythrocyte indices, including MCV, MCH, and MCHC, showed nearly identical averages between groups. WBC counts and their differential counts, including neutrophils, lymphocytes, monocytes, eosinophils, and basophils, were also similar between SCH and NCC. Monocyte and basophil counts were higher in cows with SCH, but the differences were not statistically significant. The platelet counts did not differ significantly between the two groups. Overall, although minor numerical variations were present, the CBC parameters were stable and comparable between the SCH and NCC.

### 3.6. Serum Biochemistry Parameters

Serum biochemical profiles were compared between cows diagnosed with SCH and their healthy counterparts (Table 2). Notable differences in the levels of several metabolic indicators were observed. PHOS levels were higher in SCH cows (2.28 ± 0.46 mmol/L) than in NCC (2.10 ± 0.30 mmol/L), both within the expected reference range of 1.29–2.78 mmol/L. Similarly, TP values were significantly lower in the SCH group (74.00 ± 6.60 g/L) compared to the NCC group (76.30 ± 6.50 g/L; *p* = 0.002), although both values remained within the physiological reference range (62–80 g/L).

In contrast, ALB levels were lower in SCH cows (28.20 ± 3.00 g/L) compared to NCC (29.50 ± 2.60 g/L), although both averages were within the normal range of 25–35 g/L. CHOL concentrations were also lower in SCH cows (5.22 ± 2.29 mmol/L) than in NCC (6.16 ± 2.02 mmol/L), with the average value in the NCC group exceeding the upper limit of the reference range (1.16–5.18 mmol/L). Other biochemical markers, including GLU, BHBA, Mg, BUN, \GLOB, ALB/GLOB ratio, AST, GGT, and TBIL, were numerically comparable between groups and did not demonstrate meaningful differences.

## 4. Discussion

### 4.1. Prevalence and Group Distribution

This study aimed to challenge the traditional view that SCH is restricted to the transition period. SCH refers to low blood Ca without clinical signs. Affected cows often appear outwardly healthy but are metabolically compromised. This broader pattern may reflect what has been described in the literature as persistent or intermittent SCH. Persistent SCH describes prolonged low Ca in early lactation, whereas intermittent SCH refers to recurring episodes later in lactation [18]. Although our cross-sectional design does not track individuals over time, the detection of SCH across all lactation stages, including mid and late lactation, suggests the possibility of intermittent SCH under sustained metabolic strain.

The overall individual-level prevalence of 9.4% observed in this study is lower than that reported in other countries, where the prevalence of SCH commonly ranges from 25% to over 50% in early lactation cows [1,21,22]. The lower prevalence observed here may reflect our stricter diagnostic threshold (≤8.2 mg/dL), compared to the ≤8.5 or ≤8.6 mg/dL cut-offs used in other studies. Despite the lower individual prevalence, the high herd-level prevalence indicates that SCH is widely distributed and likely endemic to the Korean dairy system. Similar patterns have been reported in other countries such as Germany and New Zealand, where studies have found that over 50% of herds have a significant proportion of cows affected by SCH [21,23]. This underscores the importance of herd-level monitoring rather than reliance on individual-level metrics alone [24].

Parity was the most significant factor associated with SCH in this study. Cows in their fourth lactation or beyond were more than three times as likely to be affected than those in their first lactation. This supports findings from large-scale studies showing that multiparous cows face a higher risk of SCH than primiparous cows, with prevalence ranging from 42% to 60% versus 11% to 25%, respectively [1,18,21,25,26]. The increased vulnerability in older cows is consistent with physiological changes reported in previous research, including impaired Ca mobilization from bone, reduced intestinal absorption, and increased Ca loss through milk [27,28,29]. Our results further suggest that these parity-related impairments persist beyond the early postpartum period, especially under sustained milk production demands [3,4,30]. This reinforces the need for Ca monitoring and support beyond the transition period, particularly in high-parity animals.

Although differences in SCH prevalence across lactation stages were not statistically significant, early lactation showed the highest proportion of affected cows, followed by mid-, high-, and late-lactation. Although statistical significance was not reached, likely due to effective Ca management or sampling variation, the presence of SCH in mid- and late-lactation cows (6–9%) remains noteworthy. These findings suggest that Ca homeostasis may remain vulnerable throughout lactation, not just during the transition phase [2,14,24]. These sporadic cases, sometimes described as “mid-lactation milk fever”, likely reflect inadequate Ca mobilization and chronic homeostatic strain in multiparous animals [21,31]. Our findings support the notion that SCH can persist beyond the transition phase. Therefore, monitoring Ca status throughout lactation, even in clinically normal cows, may help detect latent metabolic inefficiencies and prevent long-term health or productivity losses [16]. We acknowledge that the study was not designed to control for cow-level production potential within each lactation stage, and these comparisons were exploratory rather than inferential.

In this study, serum Ca concentrations among SCH-positive cows did not significantly differ by parity, suggesting that although older cows are more likely to develop SCH, the degree of hypocalcemia may not increase with age [4]. In contrast, early lactation cows exhibited the lowest serum Ca levels (7.86 ± 0.32 mg/dL; 1.96 ± 0.08 mmol/L) compared to late-lactation cows (8.11 ± 0.27 mg/dL; 2.02 ± 0.07 mmol/L), reinforcing the idea that early lactation poses the highest metabolic demand for Ca [11]. These findings further emphasize the need to evaluate Ca dynamics not just at calving, but throughout lactation, particularly in high-producing animals [2,24].

### 4.2. Milk Yield and Production Parameters

One of the most intriguing outcomes of this study was the lack of significant differences in milk yield and composition between SCH-affected and NCC. On average, cows with SCH produced virtually the same daily milk volume, fat, protein, and SNF content as normal Ca cows. This finding is somewhat unexpected due to the essential role of Ca in muscle contraction and enzymatic activity during lactation. However, previous studies have reported similar outcomes. Cows with SCH within 24 h of calving produced more milk than normocalcemic controls, particularly among third-lactation animals [10]. Similarly, cows with transient postpartum hypocalcemia had higher 15-week yields than those with stable Ca levels [11,18]. These findings suggest that high milk production may predispose cows to SCH rather than SCH reducing output. Our data support this interpretation: many SCH cows likely maintained milk synthesis by drawing on internal Ca reserves despite declining blood concentrations [32]. High-producing cows excrete substantial Ca daily, and if dietary absorption and homeostatic adaptation are insufficient, blood Ca may drop [2]. Hormonal compensation, such as increased parathyroid hormone (PTH) and vitamin D activity, may temporarily sustain balance [14,33]. These cows appear outwardly healthy and productive, but like metabolic tightrope walkers, they rely on narrow physiological margins to stay upright [10]. Although their apparent resilience allows them to sustain high output, this balance is fragile and energetically costly, potentially masking chronic metabolic stress. These findings underscore the need for broader Ca monitoring, particularly in high-yielding, multiparous cows [2,18].

### 4.3. Hematological and Serum Biochemistry Profiles

The observed serum biochemical differences between SCH and normocalcemic cows offer insight into Ca-PHOS regulation during SCH. SCH cows showed significantly higher phosphorus levels, though still within the reference range. This aligns with known physiology: when Ca levels decline, PTH stimulates bone resorption to restore Ca, concurrently releasing phosphorus into the bloodstream [34,35]. Although PTH promotes renal phosphorus excretion to prevent hyperphosphatemia, serum phosphorus may initially rise during early hypocalcemia. Our SCH cows likely reflect this early compensatory phase, where phosphorus mobilization is active but not yet severe enough to trigger renal or salivary loss. In contrast, advanced milk fever typically shows low serum phosphorus due to prolonged PTH-driven excretion [24,34]. The normal-to-elevated phosphorus levels observed in our SCH cows support this mechanism and are consistent with previous reports [15,34]. Clinically, the combination of low Ca with normal or elevated phosphorus may help differentiate SCH from clinical hypocalcemia, where both minerals are commonly depleted.

Cows with SCH also showed notable differences in protein markers and energy metabolism. Serum ALB levels were significantly lower in SCH cows, though still within the reference range. As the primary carrier of Ca, reduced ALB lowers total Ca by decreasing its protein-bound fraction, a phenomenon known as “hypoalbuminemic hypocalcemia” [36]. Although ionized Ca may be less affected, this shift holds physiological and diagnostic significance. In high-producing cows, particularly early in lactation, lower ALB likely reflects hepatic strain or NEB, as the liver prioritizes gluconeogenesis and acute-phase proteins over ALB synthesis [37]. The significantly lower CHOL levels in SCH cows further support this interpretation. Cholesterol, another liver-synthesized marker, is often suppressed during NEB or fat mobilization and may indicate impaired lipid export due to fatty liver [37,38]. The combination of hypoalbuminemia and hypocholesterolemia suggests that SCH may co-occur with deeper metabolic strain. One hypothesis is that NEB or hepatic dysfunction reduces Ca mobilization, increasing SCH risk [39,40]. Alternatively, SCH may impair rumen motility and dry matter intake, compounding NEB in a feedback loop [39]. Although causality cannot be established in this cross-sectional design, the associations support a broader view of SCH as a marker of systemic metabolic strain. This aligns with the concept of “suboptimal transition” cows, animals that appear clinically normal but exhibit multiple subclinical shifts (e.g., low Ca, low CHOL) that increase their risk of early-lactation disorders such as ketosis or metritis [18]. Persistent SCH, in particular, has been linked to higher disease risk, likely reflecting failed adaptation under metabolic pressure. Taken together, the significantly lower levels of TP and ALB, alongside reduced CHOL concentrations, suggest impaired hepatic synthetic function and a broader metabolic stress response in SCH cows. Although these markers remained within reference ranges, their consistent reduction may reflect subtle disruptions in liver function, energy metabolism, or protein turnover, common features of subclinical physiological strain during lactation. Although markers like ALB, PHOS, and TP can naturally fluctuate over time, the consistent group differences we observed, especially by parity, suggest these changes are not random. Still, because this was a single-time-point study, we cannot completely rule out normal variation as a factor. Future studies with repeated sampling are needed to confirm whether these shifts are stable and truly linked to SCH.

No significant differences were observed in hematological parameters between SCH and normocalcemic cows. All CBC indices remained within reference limits in both groups, suggesting that SCH cows were clinically stable and not affected by active infection or inflammation. If concurrent diseases such as mastitis were present, altered leukocyte profiles would be expected, yet this was not the case. These findings align with earlier reports that cows with metabolic disorders like SCH may appear healthy and maintain normal blood counts [6]. However, normal CBC does not imply intact immune function. Ca is essential for leukocyte signaling, and low Ca levels may impair neutrophil responsiveness even when cell numbers are unaffected. Experimental models show that hypocalcemia can reduce neutrophil function and increase susceptibility to uterine infections [5,7]. Even with normal CBC results, SCH cows may still have impaired immune function. Future studies should assess neutrophil activity, such as phagocytosis or oxidative burst, to better clarify this risk. Because SCH often escapes detection during routine clinical exams, targeted tests like serum Ca screening are important for early identification. Farmers should remember that seemingly healthy cows may still carry hidden imbalances that gradually affect fertility or increase disease risk [41].

### 4.4. Physiological Interpretation of SCH

To summarize the integrated mechanisms observed in this study and the supporting literature, Figure 5 presents a simplified physiological cascade underlying SCH in lactating dairy cows. It illustrates how hormonal, hepatic, and metabolic responses act to sustain milk production while masking vulnerabilities that remain undetectable through routine clinical monitoring.

### 4.5. Implications and Future Direction

Our findings highlight the importance of metabolic resilience and herd-level monitoring. High-producing cows maintained milk output despite low Ca levels, demonstrating resilience but also suggesting that they are operating near physiological limits, warranting proactive support [10]. Extending Ca supplementation beyond the periparturient period could benefit high-risk cows, those at parity ≥ 4 or with prior SCH [24]. Although current reports have focused on the immediate postpartum period [18], our findings suggest that targeted support during peak lactation may be advantageous. The association between SCH and lower ALB/CHOL levels indicates that energy and protein statuses may influence Ca regulation. Adequate energy intake supports liver function and ALB production, thereby improving the Ca-binding capacity. NEFAs may also bind to ALB and Ca in both the bloodstream and gastrointestinal tract, potentially reducing free Ca availability and interfering with transport or absorption during periods of elevated lipid mobilization. Managing health and providing a balanced diet, particularly with respect to Ca and phosphorus, are key to maintaining metabolic stability. Excessive phosphorus can blunt vitamin D activation and predispose cows to hypocalcemia, whereas PHOS deficiency may limit milk yield [34]. Although all farms fed standard lactation TMRs, phosphorus balance remains important. Herds with mid-lactation SCH should consider mineral formulations to support Ca homeostasis.

Our findings suggest that periodic Ca profiling is not limited to the periparturient period. Although early testing at 1–2 DIM remains important [18], additional sampling at the peak (60–90 DIM) and late lactation, particularly in older cows, could reveal ongoing SCH driven by nutritional gaps or excessive demand. Including Ca along with energy (NEFA and BHB) and liver function markers (CHOL and enzymes) in routine metabolic panels may enhance early herd-level detection of subclinical disorders. This approach aligns with evidence that multifactorial subclinical imbalances can undermine performance without overt illness [4]. Early detection of SCH allows for timely intervention, diet reformulation, supplementation, and management changes to bolster resilience. In summary, SCH, especially in older cows, can persist beyond early lactation and may remain clinically silent despite continued productivity, a form of metabolic resilience [10,27]. However, this stability may be fragile, and these animals may be a stressor away from decompensation. Alterations in phosphorus- and liver-related markers suggest that SCH may be part of a broader metabolic strain syndrome, rather than an isolated Ca issue. Our findings support broader mineral monitoring throughout lactation and tailored strategies for older, high-producing cows that are most at risk of hidden imbalances. Recognizing that SCH can arise at any point enables effective nutritional and managerial interventions to maintain resilience. Supporting Ca homeostasis during lactation may improve cow health, longevity, and herd productivity [42,43].

However, key metabolic regulators such as NEFA and PTH were not included in this study and, thus, were not evaluated in the statistical models. Their absence may have limited our ability to detect interactions between Ca status and energy metabolism. Future studies employing longitudinal designs may help identify key predictors of SCH and clarify how metabolic profiles evolve across lactation stages. These studies should also assess longer-term outcomes such as reduced fertility, immune suppression, or increased disease risk, which are not captured in cross-sectional designs. Although this study analyzed a large multi-herd dataset, no formal power analysis was conducted, and farm-level clustering was not accounted for. Incorporating mixed-effects models in future research may improve control for inter-farm variability in nutrition, housing, and management. In addition, advanced statistical techniques such as correlation analyses, multivariable regression, and unsupervised methods like principal component analysis (PCA) or clustering should be considered to uncover underlying metabolic patterns and improve SCH risk stratification. While our findings contribute to understanding SCH beyond the periparturient period, further investigation is needed to determine long-term impacts and optimize herd-level prevention strategies.

## 5. Conclusions

This study provides novel evidence that SCH in Korean dairy cows is not limited to the early postpartum period but may persist or recur throughout lactation, especially in older, high-producing animals. Although the individual-level prevalence was relatively low, likely due to a stricter diagnostic threshold, the high herd-level prevalence confirmed that SCH is widespread and likely endemic. Multiparous cows were significantly more affected than primiparous cows, highlighting the role of advancing parity in decreasing Ca homeostatic capacity. Although the lactation stage did not significantly affect SCH prevalence, the trend toward higher rates in early lactation and sustained occurrence in later stages underscores the importance of monitoring beyond the transition period. Despite having no observable impact on milk yield or composition, SCH cows exhibited distinct metabolic signatures, including elevated phosphorus levels and reduced ALB, TP, and CHOL levels, suggesting underlying energy and liver function challenges. Hematological profiles remained normal, reinforcing the subclinical nature of the condition and the limitations of routine diagnostics for detecting metabolically vulnerable cows. Because this study used a cross-sectional design, the long-term effects of SCH could not be assessed. Future research using longitudinal data is needed to determine whether SCH increases the risk of reproductive disorders, immune dysfunction, or disease development later in lactation. These findings support regular Ca monitoring beyond the transition period, especially during peak lactation (60–90 DIM), and targeted supplementation for cows in their fourth parity or beyond. Including Ca status alongside markers such as ALB, CHOL, and TP in herd health protocols may help identify cows at risk and improve long-term health and productivity.

## Figures and Tables

**Figure 1 vetsci-12-00495-f001:**
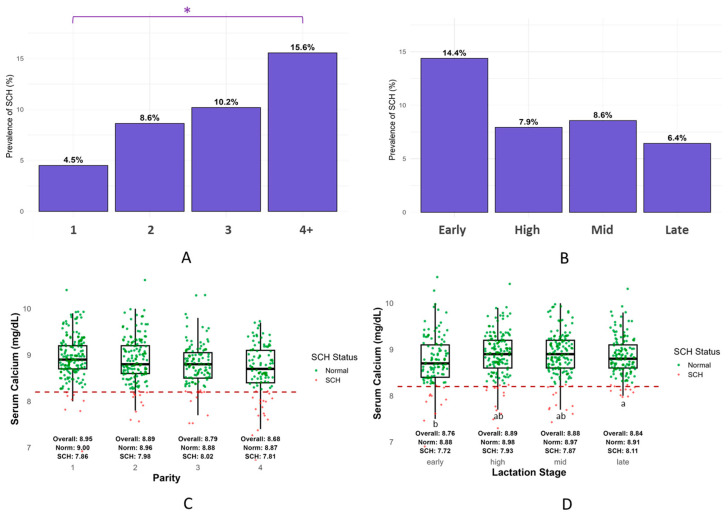
Prevalence and severity of subclinical hypocalcemia (SCH). (**A**,**B**) SCH prevalence by parity and lactation stage. Bars represent the proportion of cows within each group diagnosed with SCH. A horizontal line with (*) above bars in (**A**) indicates a statistically significant difference. Serum calcium concentrations in SCH-positive cows by parity (**C**) and lactation stage (**D**). Red dashes represent the SCH threshold (Ca = 8.2 mg/dL; 2.05 mmol/L). Letters above boxes indicate statistically significant differences (*p* < 0.05).

**Figure 2 vetsci-12-00495-f002:**
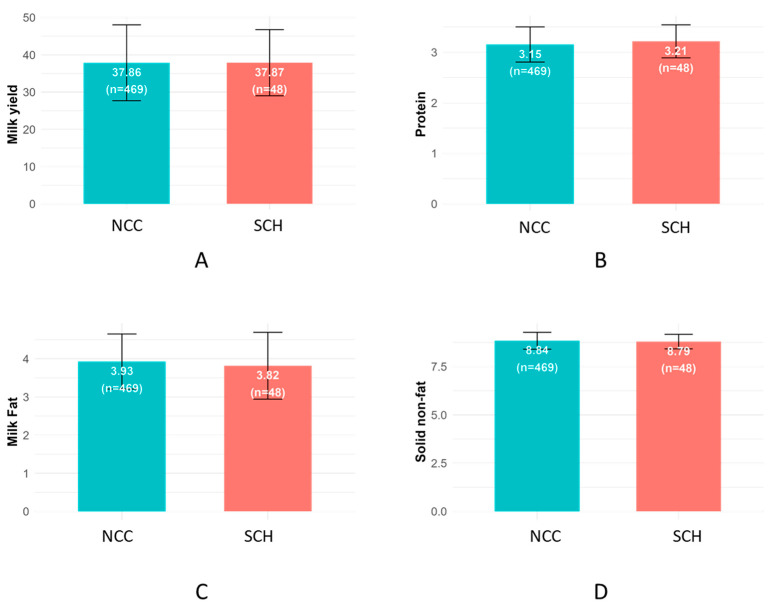
Comparison of milk parameters between normocalcemic (NCC) and subclinically hypocalcemic (SCH) cows. Bar plots represent mean ± SD for (**A**) milk yield, (**B**) protein, (**C**) milk fat, and (**D**) SNF content. No statistically significant differences were observed for any parameter.

**Figure 3 vetsci-12-00495-f003:**
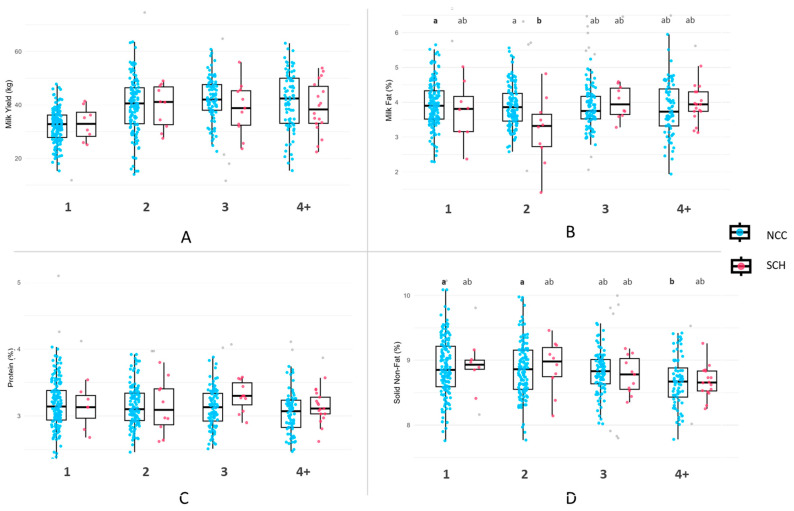
Milk yield and composition across parity groups in cows with subclinical hypocalcemia (SCH) and NCC. (**A**) milk yield (kg/day), (**B**) milk fat (%), (**C**) milk protein (%), and (**D**) solids-not-fat (%). Different letters (a, b) indicate statistically significant differences among parity groups based on two-way ANOVA. Outliers beyond 1.5× interquartile range are shown as gray dots in the boxplots.

**Figure 4 vetsci-12-00495-f004:**
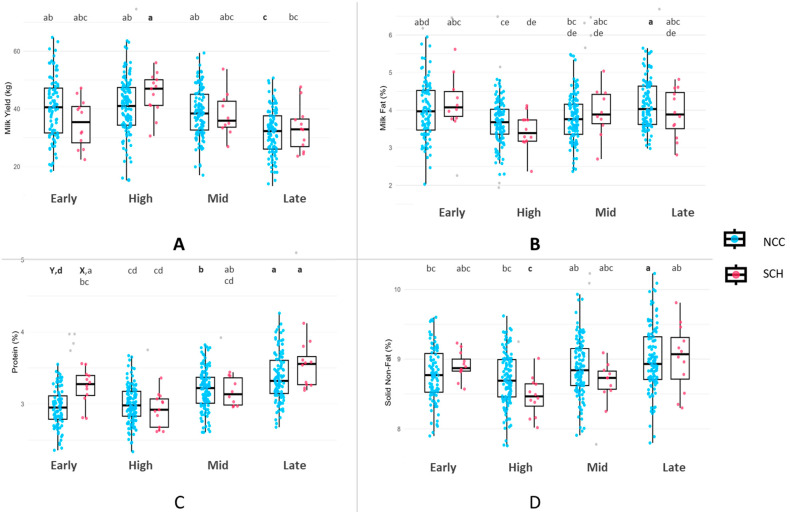
Milk yield and composition across lactation stages in cows with subclinical hypocalcemia (SCH) and NCC. (**A**) Milk yield (kg/day), (**B**) milk fat (%), (**C**) milk protein (%), and (**D**) solids-not-fat (%). Different letters (a, b, c) indicate significant differences among lactation stages, and capital letters (X, Y) denote differences between SCH and NCC based on two-way ANOVA. Outliers beyond 1.5× interquartile range are shown as gray dots in the boxplots.

**Figure 5 vetsci-12-00495-f005:**
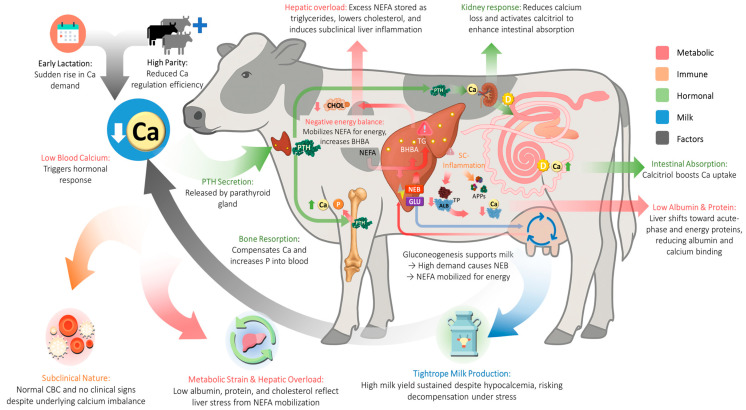
Overview of physiological and metabolic pathways involved in SCH in dairy cows. Key risk factors include early lactation, high milk yield, and advancing parity. Low blood Ca triggers PTH release, leading to bone resorption, kidney Ca conservation, and increased intestinal absorption. Negative energy balance (NEB) mobilizes NEFA for energy, but excess leads to liver overload, reduced cholesterol, and inflammation. The liver shifts protein production, lowering albumin and Ca-binding capacity. Despite these changes, cows often maintain milk yield, making SCH hard to detect through routine hematology. Arrows indicate metabolic (red), immune (orange), hormonal (green), and milk-related (blue) pathways. Grey arrows represent regulatory feedback or indirect interactions not captured by single pathways. Base cow image (with liver, bone, and kidney) generated by DALL·E; icons from Flaticon.com (CBC, Liver, and Milk Canister by Freepik).

**Table 1 vetsci-12-00495-t001:** Complete blood count (CBC) parameters (mean ± SD) in cows with subclinical hypocalcemia and NCC.

CBC Parameters	*n* (SCH/NCC)	Normal Ranges	NCC (Mean ± SD)	SCH (Mean ± SD)	*p*-Value
RBC (M/μL)	74/688	4.5–9.4	6.05 ± 0.85	5.92 ± 0.75	0.167
HCT (%)	74/688	22.5–39.9	0.3 ± 0.04	0.29 ± 0.03	0.101
HGB (g/dL)	74/688	7.4–12.8	9.94 ± 1.19	9.75 ± 1.02	0.14
MCV (fL)	74/688	40.4–56.4	49.91 ± 4.06	49.92 ± 3.53	0.974
MCH (pg)	74/688	11.5–18.5	16.57 ± 2.32	16.54 ± 1.12	0.859
MCHC (g/dL)	74/688	30.2–33.5	33.29 ± 5.36	33.15 ± 1.06	0.576
WBC (K/μL)	74/688	2.7–17.8	12.21 ± 6.04	12.32 ± 6.69	0.893
NEU (K/μL)	74/676	0.7–7.0	2.66 ± 1.94	2.55 ± 2.05	0.643
LYM (K/μL)	74/676	1.2–10.6	6.14 ± 4.13	5.96 ± 3.01	0.644
MONO (K/μL)	74/676	0.02–2.2	3.01 ± 2.21	3.45 ± 3.48	0.301
EOS (K/μL)	74/687	0.0–1.2	0.36 ± 0.33	0.35 ± 0.45	0.772
BASO (K/μL)	74/687	0.0–0.04	0.01 ± 0.02	0.02 ± 0.02	0.473
PLT (K/μL)	74/682	147.0–663.0	247.73 ± 143	245.64 ± 125	0.893

**Table 2 vetsci-12-00495-t002:** Serum biochemical parameters (mean ± SD) of subclinically hypocalcemic (SCH) and normocalcemic (NCC) cows. Reference ranges for dairy cattle are based on values provided by the IDEXX Catalyst One Chemistry Analyzer, Westbrook, ME, USA.

Biochemical Marker	n (SCH/ Normal)	Reference Range (SI)	NCC (Mean ± SD)	SCH (Mean ± SD)	*p*-Value
GLU (mmol/L)	76/726	2.50–4.20	2.74 ± 0.75	2.61 ± 0.67	0.114
BHBA (mmol/L)	81/778	<1.00	1.07 ± 0.74	1.15 ± 0.94	0.461
PHOS (mmol/L)	81/778	1.29–2.78	2.10 ± 0.30	2.28 ± 0.46	0.001
Mg (mmol/L)	81/777	0.74–1.23	0.97 ± 0.00	0.96 ± 0.01	0.493
BUN (mmol/L)	81/778	3.60–8.93	5.10 ± 1.75	5.23 ± 1.58	0.500
TP (g/L)	81/778	62.00–80.00	76.30 ± 6.50	74.00 ± 6.60	0.002
ALB (g/L)	81/778	25.00–35.00	29.50 ± 2.60	28.20 ± 3.00	<0.001
GLOB (g/L)	81/777	30.00–49.00	46.90 ± 5.00	45.70 ± 5.80	0.063
ALB/GLOB Ratio	81/777	0.80–1.20	0.63 ± 0.09	0.63 ± 0.10	0.802
AST (U/L)	81/778	50.00–150.00	112.66 ± 38.91	116.10 ± 35.79	0.416
GGT (U/L)	81/778	0.00–87.00	37.94 ± 23.49	36.14 ± 10.94	0.224
TBIL (μmol/L)	78/704	0.00–12.00	7.01 ± 5.65	6.67 ± 3.77	0.395
CHOL (mmol/L)	80/778	1.16–5.18	6.16 ± 2.02	5.22 ± 2.29	0.001

## Data Availability

The data supporting the findings of this study are available within this article.

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
