# Peer review of "Subclinical Hypocalcemia Across Lactation Stages Reflects Potential Metabolic Vulnerability in Korean Holstein Cows"

_vetsci, 2025, doi:10.3390/vetsci12050495_

Round 1
Reviewer 1 Report
Comments and Suggestions for Authors
This study reveals the persistent risk and metabolic associations of SCH during lactation in Korean dairy cows through cross-sectional analyses, which challenges the traditional view of perinatal limitations and has important theoretical and practical implications. However, methodological details need further clarification, some conclusions need more careful data support, and additional metabolic mechanism exploration and targeted recommendations for management strategies are suggested. The specific modifications are as follows:
- Blood calcium ≤8.2 mg/dL was used as the SCH criterion in the text, but the basis for the applicability of this threshold during lactation (non-perinatal period) was not clarified. It is recommended to cite the threshold criteria for non-transitional cows in recent studies, supplement the ionized calcium (iCa) data to enhance the accuracy, and clarify the blood calcium test method (total/ionized calcium) and quality control.
- Although the sample size of the methods section (n=859) was large, it was not stated whether it was based on statistical efficacy analysis. It is suggested to add the basis of sample size calculation and the details of GLMM (Generalized Linear Mixed Model) model to control the differences among herds (e.g., feed management, regional differences).
- It is recommended to add that the GLMM did not detect key metabolic regulators such as NEFA (non-esterified fatty acids) and PTH (parathyroid hormone), which may have missed evidence of calcium-energy metabolism interactions, and this needs to be clarified in the discussion.
- In the results, milk yield was not significantly decreased in SCH cows, but albumin and cholesterol were decreased and phosphorus was increased. This needs to be explained in the context of negative energy balance (NEB) and hepatic dysfunction, with relevant evidence cited.
- The CBC parameters in the results were normal but the risk of immunosuppression was not assessed. Additional neutrophil function tests (e.g., phagocytic activity) are recommended to verify the potential impact of calcium deficiency on immunity.
- The cross-sectional design had limitations that prevented the determination of causal relationships between SCH and metabolic markers. Additional longitudinal data are recommended to track changes in calcium dynamics and subsequent health outcomes (e.g., incidence of ketosis).
Author Response
We sincerely thank Reviewer 1 for their detailed and constructive evaluation of our manuscript. Your comments have helped us clarify methodological details, strengthen the interpretation of results, and better highlight the physiological implications of SCH throughout lactation. As requested, we have provided a detailed, point-by-point response to each of your comments in the attached document. We hope these revisions address your concerns and enhance the scientific quality and clarity of the manuscript.

Reviewer 2 Report
Comments and Suggestions for Authors
This manuscript aim to study the prevalence, metabolic, and risk factors for Subclinical hypocalcemia in lactating dairy cows. The result from this manuscript shows that SH has a little effect on milk yield , composition and blood parameters. Thus, results from this study only provide litter valuable information for milk production.
Author Response
We appreciate the reviewer’s time to review our manuscript, but we respectfully disagree with the conclusion that the study offers little value. It is true that no significant difference in milk production was observed between SCH and normal cows; this outcome was expected given the subclinical focus of the study.
The aim was to investigate physiological alterations associated with SCH, particularly changes in albumin, total protein, and phosphorus, which reflect early metabolic imbalance. These shifts, although not directly tied to production outcomes, are clinically meaningful and support the need for early detection and monitoring in transition cows.
Additionally, this study provides region-specific data on SCH in Korean dairy herds, an area underrepresented in current literature, and highlights differences across parity and lactation stages that may guide more tailored herd health strategies.
Yes, milk yield is a valuable endpoint, but we believe that focusing exclusively on production may overlook the broader significance of the findings. In the context of subclinical disease research, where the goal is early identification and physiological understanding, such a conclusion does not reflect the broader relevance or intent of the work.
Reviewer 3 Report
Comments and Suggestions for Authors
The paper, titled “Persistent Risk of Subclinical Hypocalcemia and Potential Metabolic Strain Across Lactation in Dairy Cows” addresses what is, in my opinion, a very important and timely topic. I found the subject matter of the article quite fascinating, and I read the manuscript with great interest – its clear the authors have put a lot of work in. The paper mostly aligns, I'd say, with the scope of the journal, maybe a bit niche but still relevant. However, I do believe that in its current form, it has several shortcomings that need to be addressed before it can be considered for publication. There are a few areas, in particular, where I think the authors could improve the manuscript, mainly around the stats and clarity of some of the figures, but I'll get into those in detail later in the review.
Introduction
- provides a good overview of the importance of studying subclinical hypocalcemia (SCH) beyond the transition period. However, it might benefit from a slightly stronger hook to really grab the reader's attention from the get-go. Perhaps a more striking statistic or a more compelling example of the economic impact of persistent SCH could be added early on.
- Also, literature review is reasonably comprehensive, but there are a few recent key studies on the long-term effects of SCH that seem to be missing. I can provide specific references if needed, but a quick search on PubMed or Google Scholar should help the authors identify them. It's important to make sure the review is absolutely up-to-date.
- objectives are clearly stated, which is great. However, perhaps a brief sentence or two outlining the overall study design in the introduction would help the reader better understand the flow of the paper.
- abbreviation SCH is used throughout the text, which is fine, but it might be helpful to spell out "subclinical hypocalcemia" the first time it appears in the abstract as well as the introduction, just to ensure clarity for all readers.
- In last paragraph, the phrase "Even if individual test results do not show large differences..." feels a bit conversational. Perhaps rephrasing to something like "Even when individual test results do not reveal marked differences..." would be more in line with the overall tone.
- There's one instance where "days in milk (DIM)" is used. It might be good to consistently use the abbreviation throughout the text or spell it out each time, for consistency.
- I suggest in the introduction make a greater overview of the problematics in the post partum period, such as ketosis (10.3389/fvets.2024.1437352, 10.3168/jds.2011-4463) due to also calving difficulties and other related problems, I suggest revise the literature searching for key words such as: calving difficulty, dairy cow, inflammation, rumination time in eminets journals for dairy science
Materials and Methods
- description of the study population and classification is generally clear. However, the rationale for choosing the specific threshold of 8.2 mg/dL for defining SCH needs a bit more elaboration. While the authors do mention it aligns with the lower limit of the physiological reference range, it would be good to discuss any potential limitations of this threshold and why it was preferred over other possible cut-offs.
- In blood sampling section, it would be helpful to provide a bit more detail on the handling and storage of the samples. For example, were the samples kept on ice in the cooler? How long was the maximum time between collection and centrifugation? These might seem like minor details, but they can be important for ensuring sample integrity.
- statistical analysis section is mostly sound, but I'm a little concerned about the use of multiple t-tests. Were any corrections for multiple comparisons applied? If not, this could increase the risk of Type I errors. The authors should clarify this point.
- In section 2.2, when describing the classification of cows based on DIM, there's a slight inconsistency in the way the ranges are presented (e.g., "0-49 DIM," "50-109 DIM," "110-219 DIM," and "2220 DIM"). It would read more smoothly if the last category was presented as "≥220 DIM" to maintain the pattern.
- In statistical analysis section, "IBP SPSSv26" is mentioned. I'm guessing this is supposed to be "IBM SPSSv26" – a quick check is probably a good idea.
- Small thing, but in several places, the degree symbol is used with a space (e.g., "-20 °C"). Standard scientific writing usually puts them together (-20°C).
Results
- are generally well-presented, and the figures are clear. However, in Figure 1, it would be helpful to add error bars to panels C and D to give the reader a better sense of the variability in the data.
- In Table 1 and 2, it would be useful to add the units for each parameter in the table heading rather than repeating them in each row. This would make the tables a bit easier to read. Also, in Table 1, there seems to be a typo in the WBC unit ("K/µL$" should probably be "K/µL").
- When discussing the serum biochemistry results, the authors state that "total protein values were slightly elevated in the SCH group... remaining within the physiological range". While technically correct, this phrasing might downplay the potential significance of this finding. Even small changes in protein levels could be biologically relevant, so I suggest the authors discuss this result with a bit more nuance.
- In Figure 1, the caption refers to "Serum calcium concentrations among SCH-positive cows..." It might be more precise to say "Serum calcium concentrations in SCH-positive cows..."
- Table 1 has a typo in the unit for WBC ("K/µL$"). It should be "K/µL". I mentioned this in the major comments too, but it's worth repeating here.
- In Table 2, some of the values seem to have inconsistent decimal places. For example, phosphorus is given as "7.06 ± 1.4 mg/dL" and "6.50 ± 0.9 mg/dL", while other parameters have only one decimal place. Maintaining consistency would improve the table's presentation.
Author Response
We sincerely thank Reviewer 3 for the thoughtful and encouraging comments regarding the relevance and value of our study. We appreciate your recognition of the work involved and helpful suggestions for improvement. In response to your comments, we have revised the manuscript to enhance clarity, refine the statistical descriptions, and address figure-related issues. A detailed, point-by-point response is provided in the attached document. We hope the revisions successfully address your concerns and strengthen the manuscript for publication.

Reviewer 4 Report
Comments and Suggestions for Authors
This manuscript offers a valuable and original contribution to veterinary science by investigating subclinical hypocalcemia (SCH) beyond the traditional periparturient period, emphasizing metabolic strain across all lactation stages in dairy cows. The study's innovative cross-sectional design, encompassing 859 cows from 49 South Korean dairy farms, and its adoption of a stricter SCH threshold (serum calcium ≤ 8.2 mg/dL) enhance diagnostic specificity. The comprehensive inclusion of biochemical, hematological, and milk production parameters provides a holistic view of SCH's impact. The study's strength lies in reframing SCH as a potentially persistent or intermittent condition, challenging the notion of it being solely a transient periparturient issue. This introduces the novel perspective that high-producing, older cows may maintain productivity while silently managing metabolic imbalances.
However, several improvements are recommended to enhance the scientific rigor and clarity. The introduction would benefit from an explicit statement of the research hypothesis to provide clearer direction for the reader. The cohort description should be enriched with details regarding diet, housing, and mineral supplementation practices to bolster the study's applicability and interpretive depth.
The title, "Persistent Risk…," needs clarification. If the aim is to identify parity, lactation stage, milk production, milk composition, and blood metabolic parameters as risk factors for SCH, a longitudinal or cohort study would be more appropriate. This study is cross-sectional. Longitudinal studies involve following the same individuals over time to observe changes or causal effects. Cross-sectional studies provide a snapshot at a single point in time, while longitudinal studies track individuals over time to observe changes or causal effects. Cohort studies follow defined groups (exposed vs. unexposed) over time to assess outcomes. This study utilizes descriptive and inferential statistics to assess SCH prevalence and its correlation with physiological and metabolic parameters. Cows were categorized into SCH and normal groups based on the serum calcium threshold, and independent t-tests were used to compare continuous variables. Chi-squared tests assessed associations between categorical variables and SCH prevalence, while two-way ANOVA evaluated interactions between SCH status and parity/lactation stage on milk production traits, using Tukey’s post hoc test for group-level differences. While no significant differences were found in milk yield or CBC values between SCH and normal cows, significant differences in several metabolic markers were observed: SCH cows had higher serum phosphorus and lower albumin, cholesterol, and total protein levels (p < 0.05), potentially indicating liver strain and energy imbalance. The study utilized a cross-sectional design and applied a combination of descriptive and inferential statistics to assess the prevalence of SCH and its association with various physiological and metabolic parameters. Cows were categorized into SCH and normal groups based on a serum calcium threshold of ≤8.2 mg/dL, and independent t-tests were used to compare continuous variables such as milk yield, blood chemistry, and hematological profiles between these two groups. Chi-squared tests assessed associations between categorical variables like parity and lactation stage with SCH prevalence, while two-way ANOVA evaluated interactions between SCH status and either parity or lactation stage on milk production traits, with Tukey’s post hoc test used to interpret group-level differences. Although no significant differences were found in milk yield or CBC values between SCH and normal cows, significant differences in several metabolic markers were observed: SCH cows had higher serum phosphorus and lower albumin, cholesterol, and total protein levels (p < 0.05), indicating potential liver strain and energy imbalance. To deepen the analysis and better understand the physiological interrelations, it is recommended that future studies incorporate Pearson or Spearman correlation coefficients to explore linear or monotonic relationships between serum calcium and individual biochemical markers, as well as multivariable linear regression models to adjust for confounders such as parity, lactation stage, and farm. Additionally, logistic regression could be used with SCH status as the outcome to identify significant predictors among metabolic markers, providing odds ratios that quantify risk associations. More advanced techniques like principal component analysis (PCA) or cluster analysis could help visualize metabolic profiles and detect patterns distinguishing SCH cows from normocalcemic animals. Collectively, these statistical approaches can enhance the understanding of how calcium metabolism interacts with broader physiological systems and inform targeted interventions in dairy herd health management.
To deepen the analysis and understand physiological interrelations, future studies should incorporate Pearson or Spearman correlations to explore relationships between serum calcium and biochemical markers, and multivariable linear regression to adjust for confounders like parity, lactation stage, and farm. Logistic regression, with SCH status as the outcome, could identify significant predictors among metabolic markers, providing odds ratios for risk associations. Advanced techniques like principal component analysis (PCA) or cluster analysis could visualize metabolic profiles and distinguish SCH cows from normocalcemic animals. These statistical approaches can enhance the understanding of how calcium metabolism interacts with broader physiological systems and inform targeted interventions in dairy herd health management.
While the study shows no significant impact of SCH on milk yield or composition, the limitations of cross-sectional data in capturing long-term outcomes should be acknowledged. Longitudinal studies are needed to explore potential hidden costs like reduced fertility or higher disease risk.
Figures and tables, while informative, should be refined for visual clarity, with simplified legends, consistent color coding, and a summary table consolidating key metabolic differences between SCH and normal cows. Figure 5 is very interesting and good.
The discussion, while rich, would benefit from clearer subheadings and reduced redundancy to improve navigation. Please refine discussion, and cut 1 to 1.5 page of discussion.
The conclusion should be more actionable by summarizing practical implications, such as: - Periodic calcium monitoring at peak lactation, - Targeted supplementation for cows in their fourth parity or beyond, - Integration of calcium with energy and liver function markers into herd health protocols.
Finally, a minor editorial revision to improve grammar, reduce passive voice, and streamline terminology would further elevate the manuscript’s professional quality.
Replace "Normal" with "Normal Ca." Convert values from mg/dL to mmol/L using SI units for all parameters. The manuscript should also address the potential roles of NEFA-albumin and NEFA-Ca complexes in blood and the gastrointestinal tract (GIT). Include analytical errors (CV%) for all blood parameters, particularly calcium, reflecting the research laboratory's established precision.
Author Response
We sincerely thank Reviewer 4 for the thoughtful and comprehensive assessment of our manuscript. We are especially grateful for your recognition of the study’s originality and its potential contributions to veterinary science. Your detailed suggestions helped us refine the framing, structure, and clarity of the manuscript, as well as improve scientific rigor. We have addressed each of your comments point-by-point in the attached response document, with corresponding revisions implemented in the manuscript. We hope these enhancements meet your expectations and strengthen the quality and interpretability of our work.

Reviewer 5 Report
Comments and Suggestions for Authors
This paper on the prevalence of subclinical hypocalcaemia in dairy cows in South Korea is an interesting piece of work that could help improve farm productivity. The paper is well structured, but the material and methods could be improved, as well as the presentation and discussion of the results. The conclusion reflects the results and the bibliography is up to date and relevant. I leave my suggestions to the authors:
Line 116: The authors should state the interval between samples and how many samples they collected.
Line 149 The authors focused on the effects of parity and SCH on milk production, but SCH is related to milk production and feed, including calcium/phosphorus intake. It would be interesting to know the effects of milk production as well, and low, medium and high production cows could be included.
Line 195-198: Authors should not state that values are higher or lower when there are no significant differences between them
Line 208: The authors found no significant differences between production and parity
Line 209: It would be convenient for the authors to put the descriptive statistics of the variables (minimum, maximum, standard deviation, number of samples) in a table
Line 232: The authors are possibly comparing animals with different production potentials, which could influence the results of the comparison between lactation stages. The authors claim to have found no significant differences ‘No statistically significant differences were observed in milk yield or composition between SCH and normal cows during the individual lactation stages, except for protein in the early stage when SCH cows showed higher levels than normal cows’ and then compare the lactation stages for differences in production, but I believe that this is not the aim of the study and it was not designed for this comparison, or was it?
Line 266: The authors should point out that the differences are statistically significant
Line 302: It's not clear what the authors mean when they say “as well as variation in herd management, nutrition, and breed productivity.”
Line 322: The authors point out that “The lactation stage was also related to a difference in SCH prevalence in our study” but they have shown that there is no relationship
Line 328: The authors point out that “and the absence of statistical significance in our results may reflect effective Ca management practices or variability in a cross-sectional design” which also applies to the parameters that have significant differences
Line 359-365: I believe the authors mention the reason in line 328
Author Response
We thank Reviewer 5 for the constructive and encouraging feedback on our manuscript. We appreciate your recognition of the study’s structure, relevance, and contribution to dairy productivity. Your detailed comments helped us improve the clarity of the Methods section, the precision of statistical interpretations, and the contextual framing of our findings. All specific suggestions have been carefully addressed in the attached point-by-point response. We trust that the revisions enhance the overall rigor and presentation of the manuscript.

Round 2
Reviewer 2 Report
Comments and Suggestions for Authors
Thank you for providing a detailed explanation. Yes, this manuscript aim to investigate physiological alterations associated with SCH, particularly changes in albumin, total protein, and phosphorus, which reflect early metabolic imbalance. In my hamble opion, these physiological indicatiors will change over time. Even sample from a same animal, the results will be differ if its sample time is changed. For example, if some one repeat this experiment, even using same animals, it migh be got divergence results. In this manuscript, the authors CAN NOT rule out these physiological alternation from normal physiological changes.
I understant some difficults in the subclinical disease research, this manuscript provide some efforts in early identification and physiological understanding in SCH. Before this manuscript can be accepted, the authors should rule out physiological alternation from normal physiological changes.
Author Response
We appreciate Reviewer 2’s thoughtful concern regarding the potential influence of normal physiological variation on serum biochemical markers. We acknowledge that markers such as albumin, total protein, and phosphorus can fluctuate over time; however, the consistent group-level differences observed, particularly across parity, suggest a non-random, biologically meaningful pattern.
We don’t intend to overinterpret isolated values, but we want to demonstrate that even within physiological ranges, subtle but consistent shifts in these markers can reflect a latent metabolic vulnerability in cows with SCH. This is especially important given the silent nature of subclinical disease and the increasing evidence that persistent or intermittent SCH can manifest as a sustained physiological burden, particularly in high-parity and early-lactation cows.
Although our cross-sectional design does not track individual animals over time, we addressed this limitation in the Discussion and have now added a clear statement (Lines 419–424) that we cannot fully rule out the role of physiological variation. However, we emphasize that the patterns seen here align with established pathophysiological mechanisms and support the view of SCH as a marker of ongoing, often overlooked metabolic strain.
We thank the reviewer for the opportunity to clarify this point, which we believe further underscores the relevance and real-world value of our findings.
Reviewer 3 Report
Comments and Suggestions for Authors
the paper improved a lot, I have no further comments
Author Response
We thank Reviewer for the positive evaluation and are grateful for the earlier constructive suggestions that helped improve the manuscript.
Reviewer 4 Report
Comments and Suggestions for Authors
Thank you very much.
Author Response
We thank the Reviewer for the thoughtful feedback throughout the review process and appreciate the final positive remarks.
Reviewer 5 Report
Comments and Suggestions for Authors
The authors made the suggested changes, successfully explaining and correcting them. I therefore consider that the paper is better and can go on to publication
Author Response
We thank the Reviewer for the clear and encouraging recommendation. We appreciate the careful review and constructive input, which strengthened the manuscript.
Round 3
Reviewer 2 Report
Comments and Suggestions for Authors
The authors provided a detailed and resonable explanation, which could addressed my major concern.
In Figure 1A, it is better to add a "*" above horizontal line.
Author Response
Response: We thank Reviewer 2 for their positive feedback and for acknowledging that the revised explanation addresses the major concern. As requested, we have added an asterisk (*) above the horizontal line in Figure 1A to clearly indicate statistical significance between parity groups. We appreciate this helpful suggestion to improve the visual clarity of the figure.